# Skeletal Muscle–Adipose Tissue–Tumor Axis: Molecular Mechanisms Linking Exercise Training in Prostate Cancer

**DOI:** 10.3390/ijms22094469

**Published:** 2021-04-25

**Authors:** Sílvia Rocha-Rodrigues, Andreia Matos, José Afonso, Miguel Mendes-Ferreira, Eduardo Abade, Eduardo Teixeira, Bruno Silva, Eugenia Murawska-Ciałowicz, Maria José Oliveira, Ricardo Ribeiro

**Affiliations:** 1Escola Superior de Desporto e Lazer, Instituto Politécnico de Viana do Castelo, Rua Escola Industrial e Comercial de Nun’Alvares, 4900-347 Viana do Castelo, Portugal; edsilvateixeira@hotmail.com (E.T.); silvabruno@esdl.ipvc.pt (B.S.); 2Research Centre in Sports Sciences, Health Sciences and Human Development (CIDESD), Quinta de Prados, Edifício Ciências de Desporto, 5001-801 Vila Real, Portugal; 3Tumour & Microenvironment Interactions Group, INEB—Institute of Biomedical Engineering, i3S-Instituto de Investigação e Inovação em Saúde, Universidade do Porto, Rua Alfredo Allen 208, 4200-153 Porto, Portugal; andreia.matos@i3s.up.pt (A.M.); mmferreira@i3s.up.pt (M.M.-F.); mariajo@ineb.up.pt (M.J.O.); ricardo.ribeiro@i3s.up.pt (R.R.); 4ICBAS-Institute of Biomedical Sciences Abel Salazar, University of Porto, R. Jorge de Viterbo Ferreira 228, 4050-313 Porto, Portugal; 5Centre for Research, Education, Innovation and Intervention in Sport, Faculty of Sport, University of Porto, Rua Dr. Plácido Costa 91, 4200-450 Porto, Portugal; jneves@fade.up.pt; 6Research Centre in Sports Sciences, Health Sciences and Human Development CIDESD, University of Institute of Maia, ISMAI, Avenida Carlos de Oliveira Campos, Castêlo da Maia, 4475-690 Maia, Portugal; eduardoabade@gmail.com; 7Faculty of Psychology, Education and Sports, Lusófona University of Porto, Rua Augusto Rosa 24, 4000-098 Porto, Portugal; 8Research Centre in Physical Activity Health and Leisure (CIAFEL), Faculty of Sport, University of Porto, Rua Dr. Plácido Costa 91, 4200-450 Porto, Portugal; 9Physiology and Biochemistry Department, University School of Physical Education, Paderewskiego Ave. 35, 51-612 Wroclaw, Poland; eugenia.murawska-cialowicz@awf.wroc.pl; 10Laboratory of Genetics and Instituto de Saúde Ambiental, Faculdade de Medicina da Universidade de Lisboa, Av. Prof. Egas Moniz, Ed. Egas Moniz, 1649-028 Lisboa, Portugal; 11Department of Clinical Pathology, Centro Hospitalar e Universitário de Coimbra, Praceta Professor Mota Pinto, 3004-561 Coimbra, Portugal

**Keywords:** physical activity, cancer, tumor microenvironment, visceral adiposity, periprostatic fat, skeletal muscle

## Abstract

Increased visceral adiposity may influence the development of prostate cancer (PCa) aggressive tumors and cancer mortality. White adipose tissue (WAT), usually referred to as periprostatic adipose tissue (PPAT), surrounds the prostatic gland and has emerged as a potential mediator of the tumor microenvironment. Exercise training (ET) induces several adaptations in both skeletal muscle and WAT. Some of these effects are mediated by ET-induced synthesis and secretion of several proteins, known as myo- and adipokines. Together, myokines and adipokines may act in an endocrine-like manner to favor communication between skeletal muscle and WAT, as they may work together to improve whole-body metabolic health. This crosstalk may constitute a potential mechanism by which ET exerts its beneficial role in the prevention and treatment of PCa-related disorders; however, this has not yet been explored. Therefore, we reviewed the current evidence on the effects of skeletal muscle–WAT–tumor crosstalk in PCa, and the potential mediators of this process to provide a better understanding of underlying ET-related mechanisms in cancer.

## 1. Introduction

Prostate cancer (PCa) has risen to be the second most frequently diagnosed cancer for males, representing 7.1% of all new cancer cases and 3.8% of all cancer deaths [1].

Existing data on the association between obesity and PCa incidence are inconsistent. However, some epidemiological evidence has demonstrated an association between obesity, specifically increased visceral adiposity, and PCa aggressiveness, poor treatment outcomes, and cancer mortality [2,3]. Considering that male subjects likely accumulate adiposity centrally, this question is of utmost importance for PCa patients [4].

White adipose tissue (WAT) is an organ closely associated with the etiology and/or pathophysiology of obesity. It produces a remarkable level of plasticity by increasing or decreasing the adipocyte cell size and number or inducing a brown-adipocyte-like phenotype through transdifferentiation processes to respond to a variety of stimuli, including diet or exercise training (ET) [5,6,7]. The WAT, usually referred to as periprostatic adipose tissue (PPAT), surrounds the prostatic gland and contributes to tumor microenvironment (TME) modulation once PCa cells invade periprostatic adipose tissue (PPAT) [8,9]. The PPAT appears to be involved in signaling pathways that influence PCa progression, emphasizing pivotal roles for WAT and obesity in PCa [10,11]. Increased local and systemic inflammation and fatty acid (FA) availability and the dysregulation of insulin and growth factor (GH), the insulin-like growth factor (IGF)-1 axis, sex steroid hormones, and adipokine synthesis and secretion [12] have emerged as possible mechanisms that could underpin the complex and intricate crosstalk between PPAT and PCa.

Skeletal muscle is a highly malleable tissue that has a notorious impact on the whole-body metabolism [13]. A loss of skeletal muscle mass and strength apparently impacts cancer survival, not only through physiologic and metabolic-related mechanisms, but also through behavioral aspects, e.g., reduced physical activity levels due to deconditioning and fatigue imposed by cancer treatments [14]. Exercise training (ET) induces several adaptations in both skeletal muscle and WAT that likely result in protective effects on metabolic and inflammatory features that typically characterize obesity and cancer diseases [5,7,15,16]. Moreover, ET promotes the synthesis and secretion of several factors known as myokines (when released by skeletal muscle) or adipokines (when released by WAT). Together, they likely act in an endocrine-dependent manner to favor the communication between skeletal muscle and WAT [17,18], as they likely work in partnership to improve whole-body health. This crosstalk may constitute a potential mechanism by which ET exerts its beneficial roles in the prevention and treatment of PCa-related disorders; however, this question has not been explored.

Therefore, in the present review, we focus on the current evidence on the effects of skeletal muscle–adipose tissue–tumor crosstalk in PCa and the potential mediators of this process in an attempt to determine the underlying ET-related mechanisms. A special box briefly describes cancer-induced cachexia in obesity (Box 1).

## 2. Prostate Cancer and Obesity

PCa is the second most frequently diagnosed cancer and the sixth most common cause of cancer-related mortality among men worldwide. Obesity has been established as a major risk factor for several cancers, including PCa [19,20]. Concerning the epidemiological association between obesity and PCa, the related body of evidence strongly suggests that excessive visceral adiposity represents a major risk factor for developing more aggressive disease and a higher PCa-specific mortality [3,21]. Nevertheless, controversies exist, particularly due to studies that have used body mass index (BMI) as a proxy for adiposity. Findings from meta-analyses [3,19,22,23] have demonstrated a weak association between BMI and PCa aggressiveness [23,24]. Considering that the BMI may also reflect increased lean fat mass (e.g., bone, muscle), other anthropometric parameters, such as waist circumference (WC) and waist–hip ratio (WHR) have been reported to be more consistent [3,25]. Both WC and WHR, in combination with BMI or not, seem to be more accurate indicators of WAT distribution [3,25].

Thus, the increase in WAT mass is responsible for the development of several obesity-related complications when white adipocyte hypertrophy is accompanied by an inadequate supply of nutrients, growth factors, and oxygen [26,27]. In this context, a set of events may occur at the WAT and other tissues in contact with it, including hypoxia, apoptosis, immunoinflammatory upregulation, extracellular matrix remodeling, and metabolic and endocrine dysregulation [28,29,30,31]. Taken together, these events might favor tumorigenesis [2]. Although the association between obesity and PCa has been supported by several studies [3,23,24,25], understanding and demonstrating the causal biological mechanisms remains unaccomplished.

### 2.1. Obesity-Induced Periprostatic Adipose Tissue Modifications

The human prostate gland is surrounded by periprostatic adipose tissue (PPAT) that lies directly adjacent to the prostatic capsule [32]. According to its anatomic localization, it is noticeable that the surrounding microenvironment of PPAT influences the prostatic gland [8,9]. Greater PPAT areas and thicknesses, which are influenced by age and body composition, have been shown to be associated with PCa aggressiveness [33,34]. Thus, in this adipocyte-rich microenvironment, namely when tumors contact the prostatic capsule, likely promotes cancer growth, adipocytes and obesity have a pivotal role in PCa tumor aggressiveness [10,11]. Extracapsular extension is a well-appreciated factor that is involved in PCa aggressiveness, local dissemination, and recurrence [35]. Therefore, the complex and intricate crosstalk between PPAT and PCa cells is likely to shape the prostate TME and might be responsible for more aggressive PCa behavior. The previously described role of PPAT in PCa reveals the prominent impact of local WAT deposits in tumor progression, even beyond global adiposity measures.

#### 2.1.1. Bidirectional Paracrine Signaling

Several different types of carcinoma are known to invade adjacent WAT, leading to crosstalk between adipocytes and malignant cells. The reciprocal interaction between cancer cells and adipocytes is highlighted by the reprogramming of adipocytes to a dedifferentiated status, termed cancer-associated adipocytes (CAAs) [11]. CAAs delipidation is accompanied by the secretion of inflammatory cytokines and proteases that fuel the rapid growth of neighboring cancer cells and prompt a tumor-permissive niche, promoting tumor growth and metastasis [11,32,36]. In PCa, periprostatic adipocytes promote the extracapsular invasion of PCa by secreting CCL7 and stimulating the migration of CCR3+ tumor cells. Obesity upregulates the secretion of CCL7, contributing to increased extraprostatic extravasation and dissemination. Accordingly, a high level of CCR3 expression in PCa is associated with extended dissemination and aggressive disease [35]. Moreover, PPAT from obese patients has been proven to secrete increased amounts of matrix metalloproteases and to promote both the survival and migration of PCa cell lines [9].

Notwithstanding, the influence of WAT on tumor initiation and progression goes beyond adipocyte-specific interactions. The adipose stromal compartment also plays a critical role in cancer development. Mesenchymal adipocyte precursors, termed adipose stromal cells (ASCs), have also been implicated in cancer. Given the rising numbers of ASCs in the PPAT of obese PCa patients [9,37], it is apparent that these cells contribute to the obesity-cancer relationship. In an obesity-driven mouse model of myc-induced PCa, the increased expression of CXCL12 from ASCs promotes the homing of CXCR4+ and CXCR7+ cells to the PPAT [38]. The CXCL12/CXCR4 axis has been implicated in the recruitment of both local adjacent and distant ASCs, showing that distant adipose depots have a critical role in PCa [39]. In addition, the inactivation of CXCL12 in ASCs was shown to preclude tumor growth and the epithelial to mesenchymal transition in Myc-driven murine PCa models [40]. Importantly, ASCs can be recruited to the TME from adjacent or distant adipose depots via chemokine gradients [41]. ASC trafficking to prostate tumors was demonstrated to be dependent on the production of CXCL1 and IL-8 to chemoattract CXCR1+ and CXCR2+ ASCs. In fact, under conditions of obesity, PCa display increased expression of CXCL1 and IL-8 chemokines, promoting the homing of ASCs to the tumor microenvironment and contributing to tumor growth promotion [42]. Infiltrating ASCs, upon arrival, contribute to a series of tumor-promoting processes that include the recruitment of macrophages and T lymphocytes to the TME and the promotion of angiogenic pathways. In fact, ASCs recruited to the tumor stroma are capable of differentiating into different cell types, including adipocytes, fibroblasts, and pericytes, depending on the environmental cues and tumor needs, providing another route for cancer progression and metastasis [41,43,44].

#### 2.1.2. Fatty Acid Availability and Metabolic Adaptations

Metabolic reprogramming is now widely considered to be a decisive process that supports the malignant properties of cancer cells [45]. In fact, tumor cells often change their metabolic status, generating adenosine triphosphate via glycolytic processes rather than oxidative phosphorylation, even under aerobic conditions, through a process known as the Warburg effect or aerobic glycolysis [45,46].

Cancer metabolism is heterogeneous and highly dependent on the TME cues and characteristics of the tissue of origin [47,48]. Importantly, unlike other solid tumors that are largely considered to be highly reliable on glycolysis, PCa metabolism is thought to be less glycolytic, while increases mitochondria oxidative phosphorylation and lipogenesis drive tumor progression [49,50]. Notably, androgen-receptor-mediated metabolic reprogramming increases the mitochondrial respiratory rate and de novo lipid synthesis [51]. However, the Warburg effect has been observed in the treatment of advanced PCa, specifically in metastatic tumors, which makes glucose-derived chemotherapeutics a promising treatment for this type of PCa [52].

It is becoming increasingly clear that the heterogeneity of cancer metabolism depends on both intrinsic cancer cell factors and other factors imposed by the tumor landscape [53]. For instance, lactate derived from glycolytic processes in hypoxic cancer cells is leveraged by well-oxygenated neighboring cancer cells to produce energy via oxidative phosphorylation, underlying metabolic symbiosis that is mainly controlled by hypoxia inducible factor (HIF)-related components through the regulation of glucose and lactate transporters in different cancer populations. [45,53,54]. On the other hand, tumors surrounding non-transformed cells may support tumor growth [55]. Notably, cancer-associated fibroblast catabolic processes “feed” anabolic cancer cells [12].

Lipid-filled adipocytes are found in the vicinity of tumors, and the reciprocal crosstalk between these cells leads to profound metabolic alterations that result in the release of free FAs from the adipocyte compartment to sustain tumor growth [56]. This metabolic symbiosis demonstrates that energy transfer occurs in cancer cells and is metabolically able to reprogram both local and distant tissues, including PPAT [12,57]. The survival-promoting effect of cancer-associated adipocytes relies on the capability of these cancer-modified adipocytes to provide nutrients to tumor cells, sustaining growth and survival and promoting metastatic behavior [58,59,60]. This impact has been evidenced by the ability of PCa cells to induce a cancer-associated adipocyte phenotype in vitro and in vivo, reflecting a more invasive phenotype [9]. This aggressive phenotype is also ensured by the induction of oxidative stress in a nicotinamide-adenine dinucleotide phosphate oxidase-dependent manner [9].

Hypoxia leads to increased lipid usage by cancer cells and lipid metabolization, which is a feature of several human cancers, including PCa [61,62,63], since it provides the necessary energy for cancer growth [64]. FAs are the main source of lipids used by cancer cells. The use of FAs to produce energy via FA oxidation increases the energetic yield compared with other energy sources [65,66]. Notably, malignant cells are capable of synthesizing FAs de novo, even in the presence of exogenous FA sources. In fact, in fast growing tumors, endogenous lipogenesis becomes limiting, and transformed cells increase their extracellular FA uptake [67]. The lipolytic state induced by cancer cells on neighboring adipocytes increases the FA availability and fulfils the energetic needs of the demanding and expanding tumor mass [36,57]. Fatty acid-binding proteins (FABPs) mediate the trafficking of adipocyte-derived FAs toward cancer cells. In fact, FABP4, mainly expressed in adipocytes, has already been shown to mediate this transport. In turn, cancer cells upregulate the expression of FA transporters (e.g., cluster of differentiation 36 (CD36)), increasing the influx of FAs and providing the tumor with sufficient substrates to sustain growth and progression [68]. In PCa, extracellular FAs seem to be crucial contributors to lipid synthesis and tumor progression [69], and the inhibition of FA uptake by CD36 abrogation has been demonstrated to be an effective strategy to reduce tumor growth in preclinical models of PCa [70]. Therefore, adipocyte-derived FAs might constitute an important substrate to support tumor growth and metastasis. Disabling FA uptake by cancer cells and targeting lipid metabolism are important ways to improve therapies and tailor novel therapeutic approaches.

#### 2.1.3. Obesity-Driven Inflammation

Inflammation is a hallmark of cancer and a critical link in obesity-driven cancer initiation and progression [43,45]. The unbalanced energy intake associated with WAT hypertrophic growth generates surplus lipid accumulation, leading to alterations in insulin sensitivity and mitochondria functioning [71]. This condition triggers a series of defective neoangiogenic mechanisms and a consequent hypoxic environment that leads to a sustained low-grade chronic inflammatory state [71,72,73]. Chronic inflammation represents a prolonged and dysfunctional protective mechanism in response to a loss of tissue homeostasis and emerges as a driving mechanism for cytotoxic accumulation of reactive oxygen and nitrogen species, leading to genomic instability [44,45,74]. In lean individuals, WAT immune infiltration maintains tissue homeostasis, contributing to the clearance of apoptotic cells, angiogenic regulation, and extracellular matrix remodeling, preserving the balance of the microenvironment. On the other hand, adipocyte death, tissue hypoxia, and mechanical stress induced by obesity [27,75] promote the recruitment and immune cell phenotypic reprogramming toward a proinflammatory profile that contributes to a sustained inflammatory state [76]. Notably, the hypoxic microenvironment induced by obesity contributes to immune cell exhaustion, increasing the expression of immune checkpoints such as programmed cell death-ligand 1 (PDL1 or CD274) and its receptor, programmed cell death protein 1 (PD1 or CD279), on tumor-associated macrophages and T cells, respectively, via HIF-1α, suppressing immunosurveillance [77,78,79].

WAT hypoxia upregulates the secretion of proinflammatory cytokines, namely tumor necrosis factor (TNF)-α, interleukin (IL)-1β, IL-6, and monocyte chemoattractant protein (MCP)-1, providing a tumor-permissive niche for transformed infiltrating cells [31,80,81]. Macrophages accumulate in the vicinity of dying adipocytes and create phagocytic immune aggregates, termed crown-like structures, and scavenge lipids and cellular debris, which additionally exacerbates systemic and local inflammatory states with a high level of production of proinflammatory cytokines [41,82,83]. Importantly, the presence of crown-like structures has been associated with a higher prostate tumor grade in men diagnosed with PCa [84]. Obesity increases the ratio of M1 pro-inflammatory to M2a anti-inflammatory macrophages by enhancing the recruitment of C-C chemokine receptor type 2^+^ pro-inflammatory macrophages [85]. Moreover, obesity contributes to the enrichment of WAT-educated helper and cytotoxic T cells through the additional aggravation of inflammatory mechanisms by secreting matrix metalloproteinases, vascular endothelial growth factor (VEGF), TNF-α, IL-6, MCP-1, and IL-1β, therefore leading to an obesity-driven inflammatory state [43,76,86,87].

This inflammatory environment helps to sustain pro-tumorigenic activities [88]. In fact, IL-1β has been shown to promote PCa progression through the repression of androgen receptor expression and promotion of the adaptor protein p62, a partner required for nuclear factor kappa B induction of cancer cell proliferation, invasion, and metastasis [89]. Convincing results have been obtained for IL-6 whose roles in PCa initiation, progression, and prognosis have been supported by several studies [88,90,91,92]. The level of IL-6 in PPAT was approximately 375 times greater than that observed in patient-matched serum. Additionally, the level was correlated with the pathological grade [92]. These findings demonstrate that there is greater phosphorylation on the signal transducer and activator of transcription 3 in high-grade tumors (any component of Gleason score 4 or 5) [92]. IL-6 has been suggested to be an oncogene in PCa that upregulates insulin growth factor (IGF) signaling [28], reprograms prostate gene expression, and drives PPAT inflammation [93].

#### 2.1.4. Adipokines

A recent study contributed an important clue to PCa pathology using liquid chromatography-mass spectrometry-based proteomic analysis in PPAT-conditioned media [94]. Some adipokines, such as IL-6, adiponectin, and chemokine (C-C motif) ligand 7, were found to be associated with PCa TME, while results for other factors are controversial. In this context, the leptin concentration of PPAT was not found to be correlated with PCa aggressiveness in a series of 30 PPAT samples obtained from patients who underwent radical prostatectomy [2]. Recent evidence suggests an association between leptin and PCa [95]. The level of adiponectin decreased in PPAT exposed to PCa-conditioned media [8], suggesting an inverse association between adiponectin and the incidence of advanced PCa. The role of adipokines on PCa requires further investigation.

Other adipocyte-secreted factors that have been pointed out, like TNF-α and VEGF, were found to be associated with high-grade PCa in a small series of 69 patients who underwent radical prostatectomy [34]. On the other hand, vascularization of the prostate crosses PPAT and favors molecular exchanges between PPAT and cancer cells [96]. The expression of osteopontin, a pro-metastatic protein involved in inflammatory and immune responses, was shown to be increased in human PPAT following PCa-conditioned media treatment [32].

#### 2.1.5. Sex Steroid Hormones

Sex steroid hormones have important roles in the function of the prostate and may also serve as a factor in the initiation and progression of carcinogenesis [29]. Obesity or the presence of a hyperinsulinemia state likely contributes to a slow decrease in estradiol levels, but this is not always to the same extent given the moderating effect of increased peripheral aromatization [30]. Obesity is a key regulator of systemic sex steroid levels through the peripheral conversion of testosterone to estrogens via aromatase in WAT [29].

Prostate epithelial cells require androgens both for normal physiology and during malignancy development. Androgen deprivation therapy (ADT) is the primary systemic treatment for hormone responsive advanced PCa, but it seems to also have an interesting role in the earlier stages of disease [97]. Men receiving ADT suffer from muscle mass loss and strength, anemia, sexual dysfunction, and accelerated bone loss and fractures [97]. Interestingly, ADT is associated with a proinflammatory and obesity-like PPAT microenvironment with adjacent effects in the PPAT [88]. In ADT-treated patients, the PPAT showed a transcriptomic pro-inflammatory signature, characterized by the up-regulation of IL-6/janus tyrosine kinase/ signal transducer and activator of transcription 3 pathway, the increase of interferon gamma response, and the increment of macrophages within PPAT [88] These findings seem to be relevant for localized disease, whereas further characterization of this fat depot in later stages of disease is lacking, particularly phenotyping the PPAT immunological environment. Cancer-induced cachexia is discussed in Box 1.

Box 1Special box—Cancer-induced cachexia.Cancer-induced cachexia (CiC) is a multifactorial syndrome characterized by a progressive loss of body weight, mainly the loss of skeletal muscle and/or WAT mass through energy imbalance [98], representing a well-known component of cancer [99]. Although PCa has the lowest prevalence of CiC and lower rates of weight loss compared with other types of cancer [100,101], it is the most commonly diagnosed cancer in males worldwide [1]. CiC has important implications on skeletal muscle quality, both in terms of tissue quantity and functionality, which leads to progressive overall functional impairment [99,102] and is responsible for more than 20% of cancer deaths [99]. Considering only body weight, CiC and obesity may seem to be opposing diseases. However, obesity also induces severe muscle depletion, known as sarcopenic obesity, which is an independent risk factor for overall survival in several types of cancer [103,104], particularly in PCa [105]. In fact, both CiC and obesity share several underlying mechanisms that lead to severe skeletal muscle atrophy [106]. Concurrent gain of WAT and loss of skeletal muscle also occur during ADT [107,108], highlighting not only the importance of maintaining the skeletal muscle quality but also the clinical significance of adiposity, irrespective of CiC.Among the therapeutic interventions available to mitigate the overall progress of CiC, ET appears to have a significant impact on the skeletal muscle–adipose tissue–tumor axis [109]. The effects of ET on the preservation of muscle mass and force in an animal model of PCa have been reported [110]. In this study, a 20-week period on the free wheel was able to prevent skeletal muscle loss by diminishing myostatin concentrations in the skeletal muscle of a transgenic mouse prostate adenocarcinoma model. Myostatin has been described to trigger catabolic activity in the skeletal muscle by downregulating protein kinase B signaling and stimulating SMAD2/3 signaling, inducing increased protein degradation and muscle wasting (for a review, see [111]). Decreased myostatin signaling in response to ET suggests that exercise not only protects against muscle loss by preventing atrophy but also promotes positive protein turnover [110]. Although the pathophysiology of sarcopenia and muscle wasting is multifaceted, skeletal muscle must be seen as a tissue of clinical relevance that can counteract the deleterious effects of both CiC and sarcopenic obesity [112].

## 3. Exercise Training as an Intervention for Obesity and Prostate Cancer

ET is one of the most powerful lifestyle strategies used to prevent and/or mitigate against becoming overweight and/or the visceral adiposity accumulation associated with obesity [113,114]. Even when body weight is not reduced, ET has a significant impact on metabolic health [114]. Accumulating evidence reveals that ET can counteract metabolic and inflammatory features usually observed in central adiposity, strengthening the metabolic relevance of WAT on whole-body adaptations to ET and revealing a promising direct target in the treatment of obesity and associated disorders [5,6,115]. In this sense, the modulation of PPAT by ET is of utmost importance for its implications in the TME of PCa patients; however, to the best of our knowledge, no research has directly addressed this question.

Studies conducted in PCa patients have shown that ET is an important strategy that can be used to mitigate PCa-related clinical features, such as lower circulating glucose and insulin levels, increased adipose tissue mass, and whole-body inflammatory markers as well as improving hormonal levels [15,16,108]. Furthermore, the impact of ET in skeletal muscle was observed by inducing phenotypic alterations mediated by transcriptional responses that occur during and following an exercise session [7]. Some genes, such as *GRIK2*, *TRAF1*, *BICC1*, and *STAG1*, are epigenetically sensitive to acute bouts of ET, maintaining an hypermethylated state for up to 22 weeks of detraining [116]. This repeated and transient increase in the expression of ET-responsive genes in skeletal muscle confers ET adaptations over time and, thus, contributes to a better understanding of the benefits widely reported in specific populations [117,118,119,120].

### 3.1. Aerobic Exercise

Aerobic exercise promotes the ability to perform moderate to vigorous ET within the abilities of the respiratory, cardiovascular, and muscular systems to integrate physiological and functional states for a prolonged period of time [121], which can be accessed via aerobic capacity [122]. A poor aerobic capacity has been associated with a markedly increased risk of premature death [123] and the development of several non-communicable diseases [124], including cancer [125]. In contrast, a high aerobic capacity is associated with a protective effect against cancer specific mortality, cancer recurrence, and/or all-cause mortality [126]. Data from prospective population-based studies [127,128] and systematic review and meta-analysis studies [129,130] have revealed an inverse association between advanced PCa risk and physical activity, demonstrating a potential protective role of regular physical activity against the risk of aggressive disease. After diagnosis, self-reported physical activity (>3 h per week at high-intensity) was found to reduce the risk of PCa-specific mortality and recurrence [131] compared with men who completed less than 1 h per week of vigorous activity. These findings suggest that physical activity counselling has a positive impact on PCa patients. Moreover, data from the Cancer of the Prostate Strategic Urologic Research Endeavor [132] showed that walking duration and total non-vigorous activity were not associated with risk of progression, but brisk walking may inhibit or delay PCa progression among men diagnosed with clinically localized PCa. In the light of these results, ET intensity may be a crucial factor in decreasing the risk of disease aggressiveness. However, others have found no relationship between ET and PCa risk or aggressiveness [133].

In a cross-sectional study, muscle mass, size, and strength were found to be compromised in PCa patients treated with ADT [134]. A 12-week cycle ergometer program performed at 65–75% of maximal oxygen consumption (V̇O_2max_) was shown to influence peripheral tissue insulin sensitivity, visceral adiposity, skeletal muscle protein content of glucose transporter type 4 (GLUT4), and total protein kinase B in both PCa patients undergoing ADT and non-exercise groups [15]. These findings suggest that the testosterone ablation in men did not constitute an impediment for peripheral tissue to increase insulin sensitivity in response to ET, which suggests that the insulin signaling cascade is a potential underlying molecular mechanism. Hence, these effects may result from exercising in general, and not from aerobic exercise specifically, as PCa patients undergoing ADT had very similar adaptations to healthy men in response to the same ET program. Furthermore, the benefits of ET extend to other aging-related comorbidities or cancer treatments side effects. For example, patients with PCa exposed to ADT were shown to have an increased risk for developing cardiovascular disease, which could be attenuated by regular ET [16].

Regular aerobic exercise induces alterations in DNA methylation patterns in the skeletal muscle, specifically with respect to genes associated with muscle growth, differentiation, and metabolism [135]. The skeletal muscle miRNA expression kinetics in response to aerobic exercise are less dependent on variables such as age, a specific ET regimen, or ET history compared with resistance exercise. However, one of the best-characterized genes that is upregulated in response to aerobic exercise and contributes to metabolic adaptation is the peroxisome proliferator-activated receptor gamma coactivator (PGC)-1α gene [136].

As recently reviewed by Neil-Sztramko et al. [137], the dosage of ET (volume, intensity and frequency) against cancer-specific mortality, recurrence, and/or all-cause mortality in PCa patients is still scarce. Few studies have appropriately reported four or five principles of ET, such as specificity, progression, overload, initial values, and diminishing returns. Therefore, a lack of attention to the principles of ET may result in underestimation of the real effect of ET and/or leave a gap in the information required for prescribing an ET program.

### 3.2. Resistance Exercise

Resistance exercise can be characterized as voluntary contractions against an external load. It is typically performed with specifically designed equipment, free weights, and/or elastic bands [138], although it can also be performed using bodyweight as resistance [139]. Resistance exercise is widely recognized as a key strategy to promote beneficial effects for the average population, such as reversing muscle loss, reducing body fat, and improving physical function and mental health [140,141]. Resistance exercise has an important role in decreasing the risk of nonfatal and total cardiovascular disease events [119]. In some specific populations, such as elderly [120], diabetic [118], or cancer patients [142], resistance exercise plays an important role as a safe and effective strategy to improve body composition as well as increasing skeletal muscle mass and strength.

Even if muscle hypertrophy and strength gains are commonly associated with each other, it should be noted that adaptations may vary according to individuals’ genetic characteristics [143] and/or the selection of programming methods [144]. Mechanical tension is a key variable in the aforementioned neuromuscular adaptations and is mainly influenced by training load and time under tension [145], triggering a series of intracellular processes that regulate gene expression, increase protein synthesis [146], and enhance cellular responses in myofibers and satellite cells [147]. Exposure to this type of regular stress, i.e., mechanical and/or metabolic tension, promotes muscle damage, an important mediator for increasing muscle mass [145]. These myotraumas initiate acute inflammatory responses and activate the production of myokines, resulting in the release of growth factors that stimulate satellite cell activity and, thus, muscle hypertrophy [145,148]. The hypertrophic response relies mainly on anaerobic glycolysis for energy production, which promotes increased secretion and production of hormonal factors as well as the activity of growth-oriented transcription factors [149].

Although DNA methylation patterns are significantly altered in exercising skeletal muscle, these alterations are of high importance for cell function under both physiological and pathological conditions [136]. When exposed to resistance exercise, the mechanistic/mammalian target of rapamycin complex 1 pathway is known as one of the central players that mediates transcriptional changes, affecting a broad variety of genes [136]. Gene expression is regulated by miRNAs—small endogenous, single stranded, non-coding RNAs that are regulated by ET. After their identification, their potential to be used as novel diagnostic and prognostic biomarkers has increased substantially, as they have been associated with improvements in markers of cardiovascular disease, cancer, diabetes, and other pathologies [150]. The degree of individual adaptations, including DNA methylation and gene expression via specific miRNAs without altering the DNA base, known as the individual’s epigenetic “signature”, is based on previous ET history and the ET regimen (aerobic or resistance) [136,151].

As mentioned in the special box, PCa patients can experience several cancer-treatment-related adverse side-effects, such as the loss of skeletal muscle mass and strength, which usually results in increased bone frailty and a high fracture risk [152]. Given that the increase of skeletal muscle mass is a well-known factor in the maintenance and enhancement of strength capacity [153], it is vital to identify muscle-hypertrophy-related mechanisms to better understand how resistance exercise induces skeletal muscle adaptations. A few studies conducted in PCa patients have provided an idea of the effects of unimodal ET interventions. In one study, men undergoing ADT performed a resistance exercise program for 20 weeks (6- to 12-repetition maximum) [154]. They experienced significant improvements in skeletal muscle strength and endurance as well as in functional tasks (e.g., chair rise, 6-m walk) without changes to fat mass, prostate-specific antigen, testosterone, growth hormone, cortisol, or hemoglobin. Since this was an observational study, it is not possible to state that the effects were specifically due to resistance exercise or to exercise in general.

In a randomized study with 58 PCa patients undergoing ADT while participating in a 16-week resistance exercise program (three weekly sessions) or usual care [155], it was observed that no significant effects on total lean body mass in either group. Nevertheless, variations in lean body mass in the upper and lower limbs reduced significantly (by 0.49 kg and 0.15 kg, respectively). In the same study, PCa patients undergoing ADT were shown to have greater strength and functional test scores (e.g., stair climbing), although no benefits on fat mass, BMI, or quality of life were found compared with the usual care group. However, the absence of a contrasting ET modality raises concerns as to whether any effects were due to resistance exercise or to exercise in general.

A systematic review and meta-analysis assessed the effects of resistance exercise on health and fitness parameters in PCa patients [142]. Generally, the majority of resistance exercise-based programs showed promising results with significant improvements in upper and lower body strength, body fat percentage, lean body mass, trunk fat mass, and cardiorespiratory parameters. It should be noted that the studies included both ADT-treated and non-ADT-treated patients.

Globally, most randomized control trials have used a combination of aerobic and resistance exercises (ranging from two to five sessions per week) and shown improvements not only in functional performance tasks (e.g., 400 m walk test) but also in metabolic parameters (e.g., diastolic blood pressure, fasting blood glucose) and body composition [156]. Conversely, these markers have shown heterogeneous responses across trials, and those effects have mostly been derived from a small subset of studies, as examined by Bourke et al. [156]. It should be noted that although these statistical differences were detected, their translation into clinically relevant improvements remains elusive.

### 3.3. High-Intensity Interval Training

High-intensity interval training (HIIT) involves repeated bouts of high-intensity effort followed by varied recovery times [121] with promising data for enhancing physical fitness. It is recognized as a time-efficient exercise modality that overcomes the barrier of lack of time [157]. Studies on feasibility, safety, and the acute and chronic responses of cancer patients to HIIT in cancer patients are still scarce. A meta-analysis showed that HIIT is an effective training type compared to moderate and continuous ET that induces similar or higher systemic and tissue changes, including effects on V̇O_2max,_ cardiac function, mitochondrial capacity, oxidative stress or inflammation, and the perception of enjoyment [158].

In forty cancer survivors (20 breast and 20 PCa survivors), two cycle ergometer HIIT protocols (10 × 1 min at peak power output (10 × 1) and 4 × 4 min at 85–95% peak HR (4 × 4)) were demonstrated to be safe strategies after the end of primary therapy [157]. Ninety-five percent of participants completed the HIIT protocol. The authors observed that the estimated energy expenditure, heart rate, and blood lactate concentration were higher in the 4 × 4 min at 85–95% peak heart rate vs. The 10 × 1 min protocol [157]. It should be noted that if a higher training stimulus is intended, the 4 × 4 protocol is more appropriate.

In PCa patients undergoing radiotherapy, short-term participation in HIIT on the cycle ergometer (8–15 × 60 s intervals at ≥85% maximal heart rate, 3 sessions per week) increased the functional exercise capacity assessed by the 6 min walk test and counteracted cancer-treatment-related fatigue compared with the usual care group [159]. Thus, HIIT may be an optimal exercise prescription for improving various aspects of health that are typically impaired in PCa patients; however, further investigation is needed.

### 3.4. Skeletal Muscle–Adipose Tissue–Tumor Axis: Molecular Mechanisms Linking Exercise Training in Prostate Cancer

The skeletal muscle may act as an energy storage site that can be used in strongly catabolic periods, such as during cancer or cancer treatments [160]. A greater skeletal muscle mass may assist in improving cancer-related outcomes [161]. Indeed, ET has been largely used as an adjunct therapy for PCa patients [16], even though the isolated effects of ET are not completely understood [162].

Beyond the general role of adiposity in the overall body composition [163], skeletal muscle can actually display excess WAT deposits, a phenomenon known as myosteatosis, which is negatively linked to health outcomes [164]. Myosteatosis has been proposed as a prognostic biomarker in patients undergoing radical cystectomy [165] and in those with endometrial cancer [166]. While a greater muscle cross-sectional area is prognostic of lower mortality rates in older patients with cancer [167], its role may vary depending on the degree of myosteatosis present [168]. Indeed, patients with similar levels of skeletal muscle mass but distinct amounts of intramuscular WAT deposits are likely to experience greater levels of toxicity in response to chemotherapy [169].

Myosteatosis is apparently derived from enhanced local secretion of inflammatory adipokines which, in turn, may be indicative of increased insulin resistance [164]. In addition, intramuscular WAT accumulation may impair blood flow within the skeletal muscle, further aggravating insulin resistance [164] and stimulating neoplasic outgrowth [170]. Accordingly, it is possible that ET improves nitric oxide bioavailability, which would improve myocardial function and peripheral blood flow [162]. In general, chronic ET may also improve endogenous antioxidant pathways [171], potentially conferring extra protection against cancer development.

Overall, ET may have distinct effects on cancer through sarcopenia and myosteatosis in breast and PCa patients. Patients who regularly exercise showed improvements in these two factors, which are independent prognostic factors of treatment outcomes [168,169]. However, greater consensus should be achieved regarding which cut-off values for classifying active or non-active subjects may be more appropriate for both health and/or specific cancer populations. Alternatively, the skeletal muscle index and density may be combined into a skeletal muscle indicator value [168], although future research is required to understand the full applications and limitations of this index. The widely recognized profits from systematic ET programs [141] with low risks and high benefits, together with the ability to increase lean muscle mass and bone density, support its usage in PCa patients undergoing ADT [162].

#### 3.4.1. Effects of Exercise Training on the Tumor Microenvironment

ET has been shown to lack an impact on tumor growth, although it has been associated with increased tumor vascularization and a shift toward reduced metastasis in an orthotopic model of murine PCa [172]. In an in vitro study, it was observed that androgen dependent LNCaP cell lines cultured in serum from exercised subjects grew less than in a control group by limiting proliferation and cell cycles [173]. In addition, exposure to post-exercise serum resulted in a >350% increase in cancer cell apoptosis through upregulation of the pro-apoptotic p53 protein.

Mice subcutaneously implanted with LAPC-4 PC were randomly assigned to voluntary wheel running or a non-intervention control group [174]. The voluntary exercised mice exhibited less tumoral necrosis, together with a ~150% increase in the microvessel density, indicating the relevant role of ET as a potential intervention to inhibit PCa growth and its underlying biologic mechanisms.

Therefore, studies assessing the optimal dosage and modality of ET in PCa patients are required [142]. In this context, eccentric exercise should be included in future research once it has been demonstrated to be a powerful ET modality in critically ill patients [175,176].

#### 3.4.2. Effects of Exercise Training on Inflammation

Clinical data regarding the effects of ET on systemic inflammation are not straightforward. In non-cancer patients, strenuous ET bouts and heavy training loads have been shown to modify T-cell immunity towards an anti-inflammatory state by increasing the numbers of resting peripheral blood type-2 and regulatory T-cells, which produce mainly the anti-inflammatory cytokines IL-4 and IL-10, respectively [177,178]. Nevertheless, 12 weeks of ET did not affect the systemic concentrations of TNF-α, IL-6, plasminogen activator inhibitor-I, leptin or adiponectin, although a reduction of visceral adiposity was observed in PCa patients receiving ADT [15]. The effects of ET on systemic inflammation possibly depend on its baseline levels and, therefore, affect people distinctly. Moreover, obese patients experience poorer outcomes during and after PCa treatments. Under these conditions, ET may be especially helpful for these patients [179].

#### 3.4.3. Effects of Exercise Training on Skeletal-Muscle-Derived Myokines and Impacts on Adipose Tissue

The metabolic disturbances typically observed in PCa patients receiving ADT suggest that skeletal muscle–adipose tissue–tumor crosstalk may represent an important opportunity to investigate the mechanisms underlying a loss of anabolic signals, sarcopenia, and frailty, and overall, they may contribute to metabolic dysregulation [97,107]. Additionally, the promyogenic anabolic interventions associated with increased skeletal muscle mass have been described as a potential strategy to improve cardiometabolic outcomes in PCa patients receiving ADT [15]. Thus far, to the best of our knowledge, this question has been poorly investigated.

Mechanistically, the contracting skeletal muscle locally stimulates several signaling pathways involved in its morphology and metabolism and produces and secretes bioactive molecules—myokines—which act in an endocrine-like fashion in distant tissues [13], such as WAT [180]. ET-associated skeletal-muscle-derived myokines may influence the metabolic phenotype of WAT by inducing a brown-adipocyte-like phenotype and thus improving FA oxidation and insulin sensitivity [7,181]. Therefore, myokines likely provide a conceptual basis to understand, at least in part, the modulatory impact of ET on WAT, e.g., the crosstalk in the skeletal muscle–WAT axis, which is of particular interest in both obesity and cancer settings. Accordingly, several studies have focused on myokines released from skeletal muscle during and immediately after ET, such as IL-6, -8, and -15, among others [7,13]. The cleaved product of fibronectin type III-domain containing 5 (FNDC5), a PGC-1α-dependent myokine, is secreted into the circulation, designed as irisin, and has the ability to initiate a brown-adipocyte-like phenotype program in WAT [18]. Undifferentiated white adipocytes treated with FNDC5 in vitro were shown to overexpress UCP1-positive cells and increase mitochondrial density and gene expression concomitantly with increases in V̇O_2_, heat loss, and energy expenditure [17]. These cells are metabolically more active, which may be advantageous in obesity and/or cancer contexts. However, to the best of our knowledge, this mechanism has been extensively reported for WAT depots but not for PPAT.

IL-6 is a proinflammatory cytokine expressed in tumors and in the stroma TME [90], but it is also a myokine prototype [182]. Indeed, circulating levels of IL-6 were found to be acutely elevated in response to ET [13]. Therefore, IL-6 has the ability to promote metabolic alterations in the skeletal muscle itself as well as in other organs, including WAT, in an endocrine manner [13]. The beneficial endocrine effects of increased IL-6 levels in response to ET on WAT have been postulated to be related to increases in the activity of adenosine monophosphate-activated protein kinase (AMPK), an energy-sensing enzyme [182,183,184]. Considering that an increase in the IL-6 concentration correlates temporally with increases in AMPK activity, some studies have suggested that IL-6 is involved in AMPK activation during the later stage of ET when the energetic state of skeletal muscle is lower [183,184]. Furthermore, the phosphorylation/activation of AMPK has been proposed to be ET intensity-dependent [7,185]. Pre-clinical studies have found that AMPK has particularly relevance in certain primary tumors, including PCa. Chippa et al. [186] reported that both androgen deprivation and hypoxia induce AMPK activation in androgen-dependent cancer cells, which is required to initiate cells to undergo autophagy, prolonging their survival; however, further research in this area is warranted.

The impact of ET on AMPK via the modulation of adiponectin levels has been shown to inhibit PCa viability in prostate (PC-3) cancer cells [187], but, intriguingly, it increases PCa cancer cell migration and the metastatic potential of human PCa cells [188]. In contrast to the well-known metabolic benefits of AMPK upregulation, the role of ET in modulating AMP activation in PCa remains elusive.

## 4. Future Directions

It would be interesting to conduct studies addressing the effects of ET alone, i.e., without any additional therapies [162]. It should be noted that a properly designed randomized controlled trial with ET-only groups contrasted with ET plus additional therapies will contribute significantly to improving the understanding of all interactions between different therapeutical modalities.

The ET dosage (volume, intensity and frequency) and the perceived individual internal load should be appropriately used but also reported and managed to understand the influence of individual responses to ET and to close the gap in information related to the prescription of effective ET programs.

Further research should address these issues to allow the guidelines for ET prescription in patients with PCa to be updated. Currently, the American Cancer Society recommends a mixture of continuous aerobic exercise for 5 days per week or aerobic exercise 2–3 times per week at a moderate intensity [189]. Although these guidelines were published in 2012, they are still the most up-to-date version provided on their website [190]. The guidelines for ET provided by the Prostate Cancer Foundation are more generic but are still based on aerobic-based activities with an emphasis on vigorous intensity whenever possible [191]. Despite the acknowledged benefits of resistance exercise for PCa patients, there are neither recommendations nor guidelines for practicing this modality [142]. Potentially, a greater role should be granted to resistance exercise, in particular, to eccentric training. Moreover, low versus high intensity exercise will likely promote differential benefits in PCa survivors, and perhaps the two should be used complementarily [179].

The effects of ET on skeletal muscle and WAT have been studied extensively, but the effects on the PPAT–tumor axis in PCa have not. Future studies should address this question to understand how ET impacts this skeletal muscle–WAT–tumor axis.

## 5. Concluding Remarks

In summary, the ET program for PCa patients was demonstrated to promote several positive adaptations in WAT by decreasing visceral adiposity and increasing skeletal mass and strength, which may be reflected as positive modulation of the TME. While scarce, the existing data seem to indicate that in cancer, particularly PCa, the skeletal muscle–WAT–tumor axis may be affected by novel ET-induced adipokines/myokines secreted by “trained” WAT and skeletal muscle (Figure 1); however, further research is needed. Moreover, this skeletal muscle–WAT–tumor axis might be an essential mechanism by which ET exerts its effects and investigation of this axis may improve the understanding of the ET-related mechanisms underlying cancer.

## Figures and Tables

**Figure 1 ijms-22-04469-f001:**
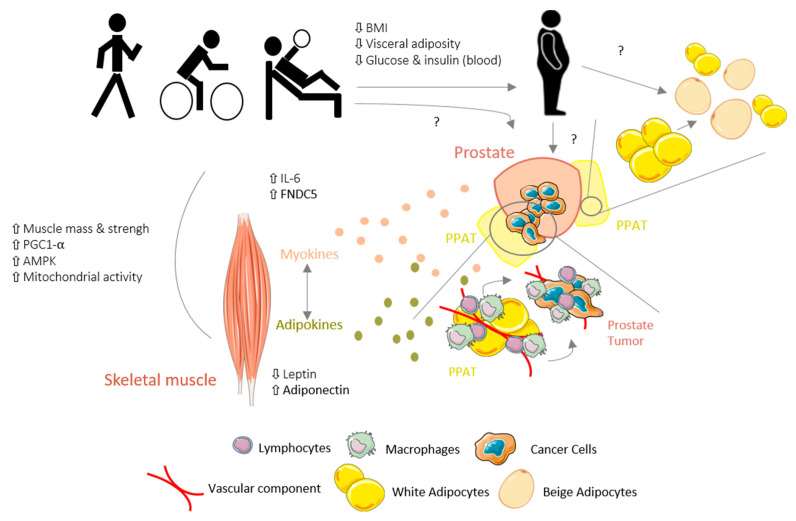
Schematic view of hypothetical mechanisms underlying the effects of exercise training (ET) on the skeletal muscle–white adipose tissue (WAT)–tumor axis in prostate cancer (PCa).ET promotes several positive adaptations in WAT by decreasing the visceral adiposity and increasing the skeletal muscle mass in PCa patients, which is reflected by a positive modulation of the tumor microenvironment (TME). Furthermore, adipokine/myokine production and secretion in response to ET may modulate the skeletal muscle–WAT–tumor axis in cancer. However, the role of ET on PPAT and PCa tumor remains elusive, with conflicting data; BW, body weight; PPAT, periprostatic adipose tissue; PGC-1α, peroxisome proliferator-activated receptor gamma coactivator; AMPK, adenosine monophosphate-activated protein kinase; IL-6, interleukin-6; FNDC5, fibronectin type III-domain containing 5.

## Data Availability

Not applicable.

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
