# Peer review of "Skeletal Muscle–Adipose Tissue–Tumor Axis: Molecular Mechanisms Linking Exercise Training in Prostate Cancer"

_ijms, 2021, doi:10.3390/ijms22094469_

Round 1
Reviewer 1 Report
It is unclear whether the authors have used the prisma guidelines in their review and what process they have used to accept publications in their review. There are moderate english language changes required for this article. Although this is an interesting article, the purpose needs to be better set up during the introduction.
Author Response
We would like to thank the careful review of our manuscript. It is our belief that your comments and suggestions certainly enriched the quality of the text and improved substantially the rationale of the discussion. All the suggestions were therefore considered and included in the revised manuscript. They are highlighted by "Track Changes" function.
Reviewer #1:
It is unclear whether the authors have used the prisma guidelines in their review and what process they have used to accept publications in their review. There are moderate english language changes required for this article. Although this is an interesting article, the purpose needs to be better set up during the introduction.
The present review is a literature review that identifies, evaluates and synthesizes the relevant literature on a research topic in a way that enables new theoretical frameworks and perspectives to emerge. So, the present review is not a systematic review. For this reason, it is not necessarily to follow the prisma guidelines.
In fact, some English language and typing error were detected after a careful final reading. All these changes were made throughout the manuscript.
Reviewer #2:
This manuscript offers some broad overview of the intricate relationship existing between solid cancers –here prostate cancer – and the adipocyte compartment of the tumour microenvironment.
Although well structured and referenced the first section on the mechanistic between PCa and the adipose tissue fails short of providing a structured synthesis on a rapidly growing literature.
We totally agree with your comments. So, a new chapter “Bidirectional paracrine signalling“ was included (please see lines 128-164) to have a structured synthesis of the literature regarding the role of PPAT in PCa.
The general impression is that of a loosely connected string of “observations” rather than that of a comprehensive understanding & presentation of a complex situation. With the objective of providing the journal audience of curious people an up-to-date introduction to a hot topic, this section would benefit from extensive restructuring. It is recommended to separate general concepts (e.g. the Warburg effect) from PCa specific data. The use of Tables and figures is suggested.
We understand your comments and suggestions, however, due to deadline of the special issue and the aim of the present review, we give an overview of the potential mechanisms linked PPAT and tumour in PCa. Notwithstanding, some sentences were added (please see lines 177-179) and a new chapter was also included (please see lines 128-164) to provide a more comprehensive understanding of this complex and intricate situation, i.e., PPAT-tumour axis in PCa.
Regarding Warburg effect, a separation between general concept and specifically in PCa was made. Please see lines 167-179.
All these suggestions were considered although we not included tables or figures, mainly due to deadlines and the aim of the review.
The section on exercise is similarly impacted by a blurring of lines between the general effects of exercise and the specific data on exercise and PCa. As a consequence, the general impression is not as sharp as would be expected from the erudition of the authors. Again, in review such as this, efforts must be paid to the clarity of the message, including the distinction between what is generally accepted “the background science” and the author’s expert understanding of the relevance of ET for PCa patients.
We understand your comments and suggestions. Some sentences and a new chapter were added (please see lines 491-514, 660-662). The relevance of ET for PCa patients and future directions were mentioned in chapter 4. Future directions.
Last, there are quite a few typos and English inadequacies in the text, which will require proof reading and correction as they often obscure the manuscript (e.g. 322-325).
The English language and typing error were rectified. All these changes were made throughout the manuscript.
Reviewer 2 Report
This manuscript offers some broad overview of the intricate relationship existing between solid cancers –here prostate cancer – and the adipocyte compartment of the tumour microenvironment.
Although well structured and referenced the first section on the mechanistic between PCa and the adipose tissue fails short of providing a structured synthesis on a rapidly growing literature.
The general impression is that of a loosely connected string of “observations” rather than that of a comprehensive understanding & presentation of a complex situation. With the objective of providing the journal audience of curious people an up-to-date introduction to a hot topic, this section would benefit from extensive restructuring. It is recommended to separate general concepts (e.g. the Warburg effect) from PCa specific data. The use of Tables and figures is suggested.
The section on exercise is similarly impacted by a blurring of lines between the general effects of exercise and the specific data on exercise and PCa. As a consequence, the general impression is not as sharp as would be expected from the erudition of the authors. Again, in review such as this, efforts must be paid to the clarity of the message, including the distinction between what is generally accepted “the background science” and the author’s expert understanding of the relevance of ET for PCa patients.
Last, there are quite a few typos and English inadequacies in the text, which will require proof reading and correction as they often obscure the manuscript (e.g. 322-325).
Author Response

(The authors gave the same response as above.)
